# Impacts of Fiber Supplementation in Sows during the Transition Period on Constipation, Farrowing Duration, Colostrum Production, and Pre-Weaning Piglet Mortality in the Free-Farrowing System

**DOI:** 10.3390/ani14060854

**Published:** 2024-03-10

**Authors:** Natchanon Dumniem, Rafa Boonprakob, Chayanat Panvichitra, Shutpisit Thongmark, Nutthawat Laohanarathip, Thanyathep Parnitvoraphoom, Siwapat Changduangjit, Tanaphum Boonmakaew, Nakarin Teshanukroh, Padet Tummaruk

**Affiliations:** 1Department of Obstetrics, Gynaecology and Reproduction, Faculty of Veterinary Science, Chulalongkorn University, Bangkok 10330, Thailand; natchanon.dum@gmail.com (N.D.); raf.boon@gmail.com (R.B.); 6135521231@student.chula.ac.th (C.P.); 6135523531@student.chula.ac.th (S.T.); 6135533831@student.chula.ac.th (N.L.); 6135547631@student.chula.ac.th (T.P.); 6135624231@student.chula.ac.th (S.C.); 6135544731@student.chula.ac.th (T.B.); 6135659231@student.chula.ac.th (N.T.); 2Livestock Production Betagro Group, Department of Quality Assurance and Animal Health Office, Bangkok 10210, Thailand; 3Centre of Excellence in Swine Reproduction, Chulalongkorn University, Bangkok 10330, Thailand

**Keywords:** constipation, farrowing, lactation, pig, transition feed

## Abstract

**Simple Summary:**

The current research evaluated the impact of adding dietary fiber to sows during the transitional phase, specifically examining its effects on constipation, the farrowing process, colostrum production, and the mortality of piglets before weaning within a free-farrowing system. Administering 75 g/day of dietary fiber to sows for 7 days before farrowing, especially in those with larger litters, showed a decrease in constipation rates and a reduction in the time taken for farrowing. Additionally, this fiber supplementation resulted in lower mortality rates in piglets prior to weaning. However, this addition of fiber did not improve the production of colostrum and milk, nor did it increase the consumption of colostrum by piglets. These results suggest that in a free-farrowing system, providing sows with dietary fiber during the transitional period can alleviate constipation and shorten the duration of farrowing. Nevertheless, it is advisable to manage the body condition of sows before farrowing to address issues like constipation effectively and to boost colostrum intake.

**Abstract:**

This study investigated how dietary fiber supplementation during the transition period in sows affects constipation, farrowing duration, colostrum production, milk yield, and pre-weaning piglet mortality in the free-farrowing system. A total of 92 Landrace × Yorkshire sows and their 1272 offspring were included in the study. Sows were divided into two groups by parity: control (*n* = 41) and treatment (*n* = 51). The control group was fed a standard lactation diet 7 days before farrowing, whereas the treatment group received the same diet with 75 g/day of a dietary fiber supplement. The dietary fiber content analyses were 4.30% and 5.53% in the control and treatment groups, respectively. Sows were fed with 3.89 ± 0.92 kg per day with a diet containing 17.8% crude protein, 3732 kcal/kg of metabolizable energy, and 1.1% lysine. Parameters including farrowing duration, litter characteristics, and the fecal condition of sows were measured. Piglet mortality rates were recorded at 3, 7, and 21 days of lactation. Sows had an average farrowing duration of 216 ± 159 min, with litter sizes of 13.8 ± 4.2 piglets per litter, and a 7.4% stillbirth rate. The treatment group showed a lower constipation rate compared to the control group (17.6% vs. 46.3%, *p* = 0.003). Farrowing duration did not significantly differ between groups, but the treatment group experienced a 43.6 min shorter duration. In sows with litter sizes of ≥16 piglets per litter, the farrowing duration for sows in the treatment group tended to be shorter than that of the control groups (202.0 ± 37.9 vs. 287.5 ± 41.3 min, *p* = 0.115). The mean colostrum intake of piglets in the control and treatment groups averaged 424.0 ± 13.7 g and 421.8 ± 12.6 g, respectively (*p* = 0.908). Likewise, the milk production of sows from Days 3 to 10 and from Days 10 to 17 of lactation in the treatment group (7.34 ± 0.27 and 8.76 ± 0.43 kg/day, respectively) did not differ from that of the control group (7.85 ± 0.30 and 9.55 ± 0.47 kg/day, respectively, *p* > 0.05). Pre-weaning piglet mortality was slightly lower in the treatment group (13.4% vs. 17.3%, *p* = 0.085). Across groups, piglet mortality rates within 3, 7, and 21 days were 10.3%, 11.9%, and 15.4%, respectively. Piglets with a ≤200 g colostrum intake had a higher pre-weaning mortality compared to those with a higher intake (*p* < 0.05), except for the 201–300 g intake (*p* = 0.472). In conclusion, introducing dietary fiber to sows during the transition period reduced constipation and tended to decrease farrowing duration, especially in sows with large litters. Additionally, it lowered pre-weaning piglet mortality by 3.9% in the free-farrowing system. Nonetheless, providing sows with a 75 g/day dietary fiber supplement for only 7 days before farrowing was not enough to enhance colostrum and milk production, nor did it lead to an increase in piglet colostrum consumption.

## 1. Introduction

In recent years, the swine industry in Thailand and many other countries has adopted group-housing and free-farrowing systems as humane alternatives for housing pregnant and nursing sows [1,2,3,4]. Studies show that sows housed in free-farrowing systems generate greater amounts of colostrum compared to those confined in crates. However, this is accompanied by elevated rates of piglet pre-weaning mortality and a higher frequency of piglet crushing incidents [2]. However, sows in free-farrowing systems have lower cortisol levels and shorter farrowing durations compared to those in crate systems [1]. The characteristics of newborn piglets and milk production are similar in both systems [2]. Our earlier study under tropical conditions found that sows with greater backfat thickness in free-farrowing systems were more likely to crush piglets than those with moderate or low backfat thickness [2]. In Europe, an increase in the body condition score from 3 to 4 in sows in free-farrowing systems resulted in longer uterine involution [3]. Furthermore, administering uterotonic agents like oxytocin or prostaglandin E2 (PGE2) in these systems did not improve the farrowing process or piglet vitality, suggesting that routine use of these agents during farrowing is discouraged [5]. These findings underscore the importance of careful management of sow body condition before farrowing in tropical environments to prevent piglet crushing in free-farrowing systems. Moreover, optimizing the free-farrowing method in both European and tropical settings is essential to support sow and piglet welfare. A quick farrowing process is crucial for piglet survival, as prolonged farrowing can increase the risk of stillbirths [6]. Factors influencing farrowing duration include pig breed, sow parity, gestation length, litter size, housing type, sow body condition, and constipation [7]. Therefore, addressing these risk factors can improve sow health and piglet survival rates.

The management of sow nutrition during the transition phase is gaining increased attention from swine production practitioners due to its substantial influence on sow productivity [4,8]. The transition phase typically spans from 5 to 7 days before parturition to 3 to 5 days postpartum, a period marked by significant physiological changes in both sows and their piglets [8]. Ensuring proper nutrition for sows during this critical time is paramount and has become a key focus for swine researchers. The widespread introduction of modern hyperprolific sows into the global swine industry, including in regions like Thailand, has been noted [9,10]. However, this has led to an observed increase in piglet pre-weaning mortality rates, raising concerns about economic losses and animal welfare. These issues may also affect sow milk yield and piglet growth rates [11,12]. Thongkhuy et al. [9] identified a link between the thickness of sow backfat at 109 days of gestation and their milk production from Days 3 to 10 of lactation, observing that every 1 mm increment in backfat thickness resulted in an additional 270 g of milk yield during this timeframe. Addressing postpartum complications in sows is crucial for reducing piglet pre-weaning mortality rates. As such, optimizing the transition feed for sows is a significant research area [13], aiming to enhance sows’ capacity to nurse large litters, reduce postpartum complications, and ultimately, improve milk production.

Constipation commonly affects sows in the later stages of gestation. A clinical study examining the prevalence of constipation in commercial swine herds under tropical conditions found that 64.6% of sows experienced moderate to severe constipation [14]. Moreover, sows with constipation had a prolonged farrowing duration by 28 min and exhibited reduced appetite post-farrowing compared to those with normal bowel movements [14]. Over the last decade, research has shown that dietary fiber supplements can alter the intestinal microbiota of sows [15,16], enhancing their intestinal motility and leading to a reduced occurrence of constipation [17]. This improvement in gut health can, in turn, enhance the performance of both sows and their litters [17,18,19]. Administering resistant starch fiber to sows before farrowing, starting from the 85th day of gestation, has been found to enhance intestinal contractions and significantly reduce the occurrence of stillbirths [18]. Additionally, both resistant starch and konjac flour have been shown to significantly decrease the production of endotoxins by intestinal bacteria, suggesting that the careful selection of dietary fibers can improve intestinal functionality and mitigate postpartum complications in sows [18]. Earlier research indicates that konjac flour is primarily composed of glucomannan, a polymer connected by β-1,4-glycosidic bonds [18]. In contrast, resistant starch, mainly consisting of Type 3 resistant starch, is produced through the gelatinization and subsequent recrystallization of amylose and amylopectin. In vitro fermentation experiments revealed that resistant starch contains higher levels of formic acid and lactate compared to konjac flour, which, conversely, exhibits higher concentrations of propionate and butyrate. Additionally, the fermentation studies showed that the populations of *Anaerovibrio* and *Erysipelatoclostridium* were more abundant in konjac flour fermentations, whereas *Proteiniclasticum* was more prevalent in those involving resistant starch [18]. These findings suggest that different fiber sources can distinctly influence the intestinal microbiome. Nonetheless, further research is necessary to explore the clinical implications of various fiber types on constipation and sow reproductive performance. This study was designed to investigate the effects of adding dietary fiber to the diet during the transition period on various aspects such as constipation, duration of farrowing, colostrum output, and the incidence of stillbirths in sows. Additionally, it evaluated lactation performance indicators such as mortality rates of piglets before weaning, weight gain across litters, and the milk production of sows. 

## 2. Materials and Methods

### 2.1. Animals and Experimental Design

This research adhered to the ethical guidelines and protocols set by the National Research Council of Thailand for conducting animal-based scientific studies. Approval was granted by the Institutional Animal Care and Use Committee (IACUC), aligning with the regulations and requirements of both the university and government concerning the care and utilization of animals in experiments (protocol number 233124, approved on 1 July 2023). The research took place within a commercial swine herd located in the central region of Thailand. The experiment included a total of 92 crossbred Landrace × Yorkshire sows, along with their offspring (*n* = 1272 piglets). The sows, on average, had parity numbers of 1.9 ± 0.7, including 24 primiparous sows, 51 sows with a parity of 2, and 17 sows with a parity of 3. The sows were divided into two groups based on their parity numbers: a control group (*n* = 41 sows) and a treatment group (*n* = 51 sows). In the control group, the sows were provided with a traditional lactation diet of 3.5–4.0 kg daily for 7 days before farrowing and 3 days after farrowing. In contrast, the sows in the treatment group were given the same amount of the diet, supplemented with a dietary fiber product of 75 g per sow per day (Lignocellulose 100%, OptiCell^®^ C5, Eubiotic Lignocellulose, manufactured by Agromed Austria GmbH, Kremsmünster, Austria). The amount of the dietary supplement given was determined by following the manufacturer’s guidelines and taking into account the fiber and energy content of the standard feed. Usually, the purpose of giving dietary fiber to sows before farrowing is to enhance digestive health and alleviate constipation, a frequent issue during the final stages of pregnancy. The precise dosage may differ depending on the fiber type, the nutritional needs of the sow, and the complete diet plan. It is typical to incrementally raise the amount of fiber as farrowing nears, to smooth the transition and promote the health of both the sow and the soon-to-be-born piglets. Table 1 displays the analysis of the chemical composition for the dietary fiber supplements. The dietary fiber content analyzed was 4.30% in the control group and 5.53% in the treatment group. The fiber supplement was provided in a dry form and mixed with the pellet-formed lactation feed before each feeding. All sows were manually fed four times a day at 6:00 A.M., 10:00 A.M., 1:00 P.M., and 4:00 P.M. The reproductive performance of sows and characteristics of piglets, along with sow feed intake, constipation, and body condition, were evaluated and compared across two groups. 

### 2.2. Sow Characteristics

The study collected various sow-related parameters, including farrowing duration, defined as the time span from the birth of the first to the last piglet. It documented the total number of piglets born in each litter (TB), the count of live-born piglets per litter (BA), the proportion of stillborn piglets per litter (SB), and the rate of mummified fetuses per litter (MF). Colostrum IgG levels in sows were measured using a Brix refractometer, employing a method validated by Hasan et al. [20]. Additionally, the fecal conditions of sows were monitored for 3 days before and after parturition, with those having an average fecal score of 2 or below classified as constipated. The study also tracked the colostrum and milk production of sows during Days 3–10 and 10–17 of lactation, to facilitate comparison between the control and treatment groups.

The researchers carefully evaluated the quantity of sows’ feed consumption from one week before farrowing until 3 days after farrowing. The sows were fed on a schedule four times a day at 06:30 AM, 10:00 AM, 01:00 PM, and 04:00 PM. Two hours post-feeding, any uneaten feed was gathered, weighed, and discarded. The consumption for each meal was calculated by subtracting the amount of remaining feed from the initial quantity provided to the sows. The cumulative daily feed intake was determined by summing up the intake across all meals. To assess constipation severity, a scoring system according to Oliviero et al. [7] was used, namely 0 indicated very severe constipation, 1 indicated severe constipation, 2 indicated moderate constipation, 3 indicated normal feces, 4 indicated fairly soft feces, and 5 indicated very soft feces. Sows were categorized as constipated if their average fecal score was 2 or lower. Furthermore, backfat thickness was assessed at 109 ± 2 days of gestation and at 21 days of lactation using A-mode ultrasonography (Renco Lean-Meater^®^, Minneapolis, MN, USA). The percentage of backfat loss for each sow was calculated.

### 2.3. Piglet Characteristics

Piglets’ data included individual birth weight, piglet birth interval, and cumulative birth interval. Newborn piglets were weighed immediately after birth using a digital scale (SDS^®^ IDS701-CSERIES, SDS, Yangzhou, Digital Scale Co. Ltd., Yangzhou, China) and numbering using a marker pen and medical adhesive tape. Piglets’ bodyweight was remeasured at 24 h postpartum to calculate weight gain and colostrum intake. Colostrum intake (CI) for each piglet was calculated according by Theil et al. [21]: colostrum intake (g) = −106 + 2.26 WG + 200 BWB + 0.111D − 1414 WG/D + 0.0182 WG/BWB, where WG represents the piglet’s weight gain over 24 h (g), BWB represents the piglet’s birth weight (kg), and D represents the duration of colostrum sucking (the time from birth to weighing at 24 h, measured in min). The sow colostrum yield was calculated by a summation of the colostrum intakes of each individual piglet within the same litter. Furthermore, the piglets were categorized into two groups based on their colostrum consumption. Briefly, piglets that had a colostrum intake <300 g and ≥300 g were classified as inadequate and adequate colostrum intake, respectively [22]. The estimation of sow milk production was calculated using an equation by Hansen et al. [23]: milk yield between Day 3 and 10 (g) = 1.93 + 0.07 × (litter size − 9.5) + 0.04 × (litter gain, kg per day − 2.05) and milk yield between Day 10 and 17 (g) = 2.23 + 0.05 × (litter size − 9.5) + 0.23 × (litter gain, kg per day − 2.05). The number of piglet deaths at 3, 7, and 21 days of lactation was recorded, and the pre-weaning mortality rates for each time point were calculated.

### 2.4. General Management and Farrowing Supervision

A conventional evaporative cooling system was implemented to accommodate the sows’ and gilts’ indoor environment. Initially, from the third day post-insemination until approximately 109 ± 2.0 days into their gestation, sows and gilts were housed collectively. They were then transferred to free-farrowing pens, which were equipped with an adjustable swing hinge and plastic slatted flooring. The dimensions of each farrowing pen were 2.00 × 2.35 × 0.90 m, offering a total space of 4.7 m^2^ per pen. On the farrowing day and for the following 3 days, the metal swing hinge was secured, confining the sows in individual crates sized 1.80 × 0.60 × 0.90 m, thus allocating 1.08 m^2^ of space per sow. Starting from the fourth day post-farrowing up to weaning, the swing hinge was unlocked and fully opened. The designated area for piglets included a heating lamp, a rubber mattress, and a feeding bowl. Throughout the study, the barn maintained an average daily temperature of 26.7 ± 0.4 °C, fluctuating between 24.7 °C and 29.2 °C. The average daily humidity in the barn was recorded at 67.0 ± 2.0%, with a variation range of 62.0% to 73.7%.

At various stages of gestation, the feed amount for each sow was adjusted daily. Initially, during the early, middle, and final stages of gestation, sows were fed 3.0–3.5 kg daily. The gestational feed comprised 12.7% crude protein, 2700 kcal/kg of metabolizable energy, 5.7% fiber, and 0.7% lysine. Upon moving to the farrowing house, sows switched to a lactation diet. In the week preceding farrowing, sows were gradually fed 3.0–3.5 kg of the lactation diet per day. This diet included 17.8% crude protein, 3732 kcal/kg of metabolizable energy, 4.3% fiber, and 1.1% lysine. The calculated composition of the control and experimental diets in this study (as-fed basis) is presented in Table 2. Feeding occurred four times a day at set times: 06:00 AM, 10:00 AM, 01:00 PM, and 04:00 PM, with the feed presented in pellet form. Post-farrowing, sows had unlimited access to feed. An automatic feeder provided unrestricted access to the lactation diet for nursing sows, leading to an average intake of 5.0–6.0 kg per sow per day. Additionally, water was freely available via drinking nipples.

The researcher carefully observed the farrowing procedure continuously for 24 h. Data on farrowing were meticulously recorded right after the birth of each piglet, encompassing the onset and conclusion of farrowing, the birth weights of the viable piglets, and their initial conditions (alive, stillborn, or mummified). Interventions during farrowing were limited to cases of dystocia, which involved manual piglet extraction and the intramuscular injection of 20 IU of oxytocin (from Phenix Pharmaceuticals N.V. Co. Ltd., Hoogstraten, Belgium) should the time between piglet deliveries exceed 60 min. Moreover, following the birth of the 10th piglet, a routine 20 IU oxytocin injection was administered to all sows to facilitate placental expulsion and initiate milk flow. Post-farrowing, all sows received an antipyretic treatment with ketoprofen (3.0 mg/kg, intramuscularly, from KELA N.V., Hoogstraten, Belgium). The herd veterinarian oversaw the health of the sows, which included vaccinations against foot and mouth disease (AFTOPOR^®^, Merial SAS, Lyon, France) and Aujeszky’s disease virus (Porcilis^®^ Ad Begonia, Merck Animal Health, Madison, WI, USA) prior to farrowing. Postpartum, vaccinations were administered for classical swine fever (from Ceva-Phylaxia Veterinary Biologicals Co. Ltd., Budapest, Hungary) and a combination vaccine for Porcine Parvovirus, Leptospira, and Erysipelas (Eryseng^®^, Laboratorios Hipra, S.A., Amer, Girona, Spain). The piglets received a vaccination against *Mycoplasma hyopneumoniae* (Hyogen^®^, Ceva Santé Animale S.A, Libourne, France) between the ages of 18 to 22 days.

### 2.5. Statistical Analysis

All statistical analyses were carried out using SAS version 9.4 (SAS Institute Inc., Cary, NC, USA). Descriptive statistics were determined for continuous and categorical variables using the MEAN and FREQ procedures. General linear models (GLM) were performed to determine the difference between treatment groups for continuous variables including TB, BA, SB, MM, farrowing duration, daily feed intake, colostrum Brix value (%), colostrum yield, milk yield from 3–10 days of lactation, milk yield from 10–17 days of lactation, backfat thickness at 109 days of gestation, and backfat loss during lactation. The statistical models included dietary fiber supplementation (control and treatment), sow parity (primiparous and multiparous), TB classes (<16 and ≥16 piglets per litter), and the interactions between the treatment group and sow parity and between the treatment group and TB classes as fixed effects. Additionally, to balance the backfat thickness of sows between groups, the backfat measurements taken at 109 days of gestation were included in the statistical model as covariates. Least-square means were calculated for each variable and subsequently compared using the least significant difference test. Additionally, Pearson’s correlation analyses were performed to determine the relationship between backfat thickness at 109 days of gestation and backfat thickness at weaning, as well as backfat loss during lactation. The generalized linear mixed models (GLIMMIX) were carried out to analyze categorical traits including the percentage of sows who experienced prolong farrowing (<300 and ≥300 min) and constipation (yes, no). Additionally, a comparison of the fecal scores of sows between the control and treatment groups was analyzed using the Wilcoxon rank sum test.

General linear mixed models (MIXED) were utilized to analyze the piglets’ data including piglet birth weight, piglet body weight at 24 h postpartum, body weight gain, colostrum intake, piglet birth interval, cumulative birth interval, and piglet pre-weaning mortality rate at 3, 7, and 21 days of lactation. The statistical models included dietary fiber supplementation (control and treatment), sow parity (primiparous and multiparous), piglet birth weight categories (<1.0 kg, 1.0–1.29 kg, and ≥1.3 kg), and the interaction between treatment groups and sow parity and between the treatment groups and piglet birth weight categories as fixed effects. To adjust for repeat measurement, the sow’s identity was included in the model as a random effect. Least-square means were calculated for each variable and subsequently compared using the least significant difference test. 

For categorical data, including the proportion of piglets that were born with SB birth status (yes/no), the proportion of piglets with colostrum intake below 300 g (yes/no), and the percentage of piglet mortality at 3, 7, and 21 days of lactation, analysis was conducted using the GLIMMIX procedure. The statistical models included dietary fiber supplementation (control and treatment), sow parity (primiparous and multiparous), piglet birth weight categories (<1.0 kg, 1.0–1.29 kg, and ≥1.3 kg), and the interactions between treatment groups and sow parity, as well as between treatment groups and piglet birth weight categories, as fixed effects. To adjust for repeated measurements, the sow’s identity was included as a random effect in the model. Least-square means were calculated for each variable and subsequently compared using the least significant difference test. Furthermore, to analyze the influence of piglet colostrum intake on piglet mortality (yes/no), the GLIMMIX procedure was performed and included colostrum intake classes (≥200, 201–300, 301–400, 401–500, 501–600, and >600 g) as a fixed effect and sow identities as random effects. For all statistical models, a *p* value less than 0.05 was considered statistically significant.

## 3. Results

### 3.1. Descriptive Statistics

The descriptive statistics for the reproductive performance of all sows are shown in Table 3. The average farrowing duration, TB, BA, and SB were 216 ± 159 min, 13.8 ± 4.2, 11.8 ± 4.0, and 7.4%, respectively, and 39.1% of the sows had a TB ≥ 16. Across groups, the average daily feed intake of sows was 3.89 ± 0.92 kg per sow. The average colostrum yield of sows was 5.26 ± 1.08 kg and varied among sows from 2.40 to 7.85 kg (Table 3). The frequency distribution of colostrum yield in sows is presented in Figure 1A. Across the groups, the milk yield of sows from 3–10 days and from 10–17 days of lactation averaged 7.6 ± 1.6 and 9.2 ± 2.5 kg per day, respectively. The frequency distributions of the milk yield in sows from 3–10 days and from 10–17 days of lactation are presented in Figure 1B and Figure 1C, respectively. Regarding the piglets, the average piglet birth weight and body weight at 24 h postpartum were 1330 ± 328 and 1451 ± 367 g, respectively. The average colostrum consumption was 450.8 ± 160.9 g. The cumulative birth interval averaged 94.9 ± 98.8 min (Table 3).

### 3.2. Sow Characteristics

The reproductive performances of sows in the control group and the treatment group are presented in Table 4. Litter characteristics were similar across the groups (Table 4). However, the treatment group showed a lower percentage of sows experiencing constipation compared to the control group (17.6% vs. 46.3%, respectively; *p* = 0.003, Table 4). While the duration of farrowing did not significantly differ between the groups, sows in the treatment group had a farrowing duration that was, on average, 43.6 min shorter (Table 4). For sows with litter sizes of 16 piglets or more, the farrowing duration tended to be shorter in the treatment group than in the control group (202.0 ± 37.9 min vs. 287.5 ± 41.3 min, respectively; *p* = 0.115).

Across groups, the mean backfat thicknesses at 109 days of gestation and at weaning were 21.6 ± 3.9 mm and 14.8 ± 3.1 mm, respectively. On average, sows lost 30.8 ± 13.0% of their backfat during lactation (Table 3). Backfat thickness at 109 days of gestation was positively correlated with both backfat thickness at weaning (r = 0.589, *p* < 0.001) and the percentage of backfat loss during lactation (r = 0.363, *p* < 0.001). The backfat thickness at 109 days of gestation was greater in the treatment group than in the control group (Table 4). Similarly, the backfat thickness at weaning was lower and the percentage of backfat loss during lactation was higher in the treatment group compared to the control group (Table 4).

### 3.3. Piglet Characteristics

Table 5 presents the piglet performance and colostrum intake for the control and treatment groups. The average colostrum intake of piglets in the control and treatment groups was 424.0 ± 13.7 g and 421.8 ± 12.6 g, respectively. The percentage of piglets with insufficient colostrum intake did not differ significantly between the control and treatment groups (*p* > 0.05).

On average, piglets weighing ≥1.3 kg at birth consumed more colostrum than piglets born weighing between 1.0 and 1.29 kg and those weighing <1.0 kg (534.2 ± 9.8 g vs. 431.6 ± 11.0 g and 302.9 ± 13.3 g, respectively; *p* < 0.001) (Figure 2A). Additionally, the proportion of piglets with inadequate colostrum intake was lower among those with a birth weight of ≥1.3 kg compared to piglets with birth weights of 1.0–1.29 kg and <1.0 kg (7.8% vs. 19.6% and 41.6%, respectively; *p* < 0.001) (Figure 2B). The piglet mortality rates during the first 3 days of life were 8.2%, 9.1%, and 19.6% for those with birth weights of ≥1.3 kg, 1.0–1.29 kg, and <1.0 kg, respectively (*p* < 0.001). Similarly, the piglet mortality rates during the first 7 days of life were 9.7%, 10.0%, and 21.8% for those with birth weights of ≥1.3 kg, 1.0–1.29 kg, and <1.0 kg, respectively (*p* < 0.001). Colostrum intake and the proportion of piglets with a colostrum intake of <300 g by birth weight in the control and treatment groups are presented in Figure 3A and Figure 3B, respectively.

### 3.4. Piglet Pre-Weaning Mortality

The pre-weaning mortality of piglets at 21 days of life was slightly lower in the treatment group than in the control group (13.4% vs. 17.3%, respectively; *p* = 0.085) (Table 5). Among the piglets that died within the first 21 days of life (*n* = 168), 112 (66.7%) died within the first 3 days, and 130 (77.4%) died within the first 7 days. Figure 4 illustrates the proportions of piglet mortality within the first 3, 7, and 21 days of life, categorized by their colostrum intake within the first 24 h postpartum. Piglets with a colostrum intake of ≤200 g within the first 24 h postpartum had a higher pre-weaning mortality rate compared to those with intakes of 301–400, 401–500, 501–600, and >600 g (21.0% vs. 8.8%, 6.6%, 7.2%, and 5.9%, respectively; *p* < 0.05). However, this rate did not differ significantly from piglets with an intake of 201–300 g of colostrum (15.5%; *p* = 0.472).

### 3.5. Sow Colostrum Yield and Milk Yield

No effect of dietary fiber treatment was observed on sow colostrum yield and milk yield (Table 4). However, colostrum yield and milk yield in sows were influenced by TB (*p* < 0.05). Sows with a TB ≥ 16 had higher colostrum production compared to those with <16 piglets per litter (5.57 ± 0.19 kg vs. 5.03 ± 0.17 kg, respectively; *p* = 0.026). Additionally, sows with a TB ≥16 had higher milk yields during the periods of 3–10 days (8.03 ± 0.29 kg vs. 7.16 ± 0.23 kg, respectively; *p* = 0.015) and 10–17 days of lactation (9.69 ± 0.46 kg vs. 8.62 ± 0.37 kg, respectively; *p* = 0.058) compared to sows with <16 piglets per litter.

## 4. Discussion

The present study underscored the importance of dietary fiber content in feed during the transition period for sows, with a focus on its effects on the occurrence of constipation, farrowing duration, colostrum production, and the incidence of stillbirths. Additionally, it explored lactation performance parameters, including piglet pre-weaning mortality, piglet body weight gain, and milk yield in sows. In the absence of additional dietary fiber, 46.3% (*n* = 19) of the sows exhibited moderate to severe constipation symptoms. Moreover, the standard lactation diet for the herd in this study contained only 4.3% fiber, markedly lower than the fiber content reported in similar European studies [19,24]. After supplementing the sows’ diet with dietary fiber during a 7-day transition period, the incidence of constipation decreased to 17.6% (*n* = 9). Despite this reduction, nine sows in the treatment group continued to experience constipation, indicating a potential need for increased dietary fiber or an alternative type of fiber [18]. Notably, the dietary fiber used in this study was predominantly composed of 86.7% insoluble fiber and only 1.3% soluble fiber. A recent study by Montoya et al. [16] emphasized the critical role of both the type and amount of dietary fiber in generating organic acids in the hindgut, a process heavily reliant on fermentable dietary components. Therefore, supplementing dietary fiber during the transition period presents complexities, potentially affecting gut microbial responses to the specific type or amount of fiber. In this study, a 7-day dietary fiber supplementation before farrowing significantly reduced constipation rates and led to a non-significant reduction in farrowing duration by 43.6 min. There was also a 3.9% decrease in pre-weaning piglet mortality. However, the supplementation did not significantly enhance colostrum production or milk yield in sows, indicating that factors other than fiber intake might play a crucial role in optimizing farrowing processes and piglet health. Future research should explore these elements in conjunction with dietary fiber supplementation.

### 4.1. Constipation

In the present study, the introduction of dietary fiber supplementation during the transition periods led to a decrease in the prevalence of constipation from 46.3% to 17.6%. This result highlights the effectiveness of incorporating dietary fiber supplementation during the transition phase for sows. Constipation is linked to inadequate water consumption and can occasionally have negative impacts on the well-being and productivity of sows. In the postpartum period, most sows exhibited reduced water intake and elevated levels of dry matter content in their feces. In hot climates, approximately 64.6% of gestating sows display symptoms ranging from moderate to severe constipation [14]. Constipation also increased the release and uptake of bacterial endotoxins, potentially giving rise to the onset of postpartum dysgalactia syndrome in sows [25]. Oliviero et al. [7] revealed that sows experiencing constipation had prolonged farrowing periods, lasting more than 300 min. These findings emphasize the significance of addressing constipation issues in peripartum sows as they are primary and crucial factors contributing to the development of postpartum dysgalactia syndrome in sows, subsequently leading to elevated mortality rates among piglets prior to weaning. Björkman et al. [26] proposed that the most effective strategies for decreasing the incidence of postpartum dysgalactia syndrome involve optimizing animal healthcare, feeding practices, and overall management. This includes the use of a prepartal transition feed containing expandable raw fiber to prevent constipation. Additionally, the administration of nonsteroidal anti-inflammatory drugs and oxytocin is also recommended in cases of postpartum dysgalactia syndrome [27].

### 4.2. Farrowing Duration

In the present study, 20% of sows experienced a prolonged farrowing duration of more than 300 min. Although the farrowing duration did not differ significantly between the control and treatment groups, sows that received fiber supplementation for 7 days before farrowing had a farrowing duration that was 43.6 min shorter than that of the control sows, which represents a clinically significant difference. Similarly, Pearodwong et al. [14] demonstrated that sows suffering from moderate to very severe constipation experienced an extension in farrowing duration by 28 min compared to sows with normal feces. Additionally, constipation in sows on the day of farrowing led to a reduced feed appetite on the first day after farrowing and an increased likelihood of sows developing fever on the same day compared to sows without constipation [14]. These findings suggest that decreasing the farrowing duration could potentially enhance the well-being of sows during the postpartum period. Furthermore, no significant difference was observed in either the piglet birth interval (16.6 min) or the cumulative birth interval (94.9 min) between the control and treatment groups. This implies that dietary fiber supplementation did not affect the physiological motility of the uterus, and therefore, the duration of uterine contractions remained unchanged. Nonetheless, the piglet birth interval and cumulative birth interval are significant biological markers for stillbirth rates and the piglets’ ability to consume colostrum [28,29].

### 4.3. Reproductive Performances

Despite extensive research investigating the positive impacts of dietary fiber from the standpoint of behavior and welfare, there is a notable scarcity of data concerning its influence on reproductive performance [30]. A study in Denmark demonstrated that adding dietary fiber to the diets of sows during the last 2 weeks of gestation led to a decrease in the percentage of stillborn piglets from 8.8% to 6.6% and a reduction in the mortality rate of all born piglets from 22.3% to 19.9% [30]. In the present study, the percentage of stillborn piglets was 7.4%, with no significant difference observed between the control and treatment groups. This outcome may be attributed to the fact that the average litter size in this study (13.8 piglets per litter) was notably lower than that in the Danish study (18.1–18.4 piglets per litter) [30]. The incidence of stillbirths is closely related to litter size, with a higher prevalence of stillbirths commonly noted after the delivery of the 12th piglet [28]. In this study, the expected benefit of dietary fiber in reducing stillbirth rates was not observed. It is important to note that the sows included in this study were relatively young, ranging from parity one to three, possibly leading to a lower probability of uterine inertia during the farrowing process compared to older sows [6,31]. Wongwaipisitkul et al. [31] demonstrated that sows in their first and second to fourth parities required farrowing assistance less frequently compared to sows with a parity greater than or equal to seven. However, within the scope of this study, piglet pre-weaning mortality was generally lower in sows receiving dietary fiber supplementation compared to the control group, with percentages of 13.4% and 17.3%, respectively, indicating a reduction of 3.9%. This suggests a beneficial impact of dietary fiber supplementation on sows. In gestating sows, providing sugar beet pulp as a fermentable dietary fiber leads to an increase in both body weight gain and body protein retention, indicating that gestating sows have a strong ability to extract energy from fermentable fiber [32]. However, it is noteworthy that sows tend to prioritize the retention of protein over fat, necessitating careful consideration when formulating nutrition for modern-genotype sows [32]. During lactation, highly prolific sows mobilize a significant amount of body fat for milk production. Consequently, it is crucial to replenish both body fat and protein lost during the subsequent gestation period. Therefore, considering the timing and dosage of dietary fiber supplementation within the reproductive cycle is important to enhance reproductive performance and preserve body condition. Moreover, in this study, the treatment group exhibited a greater percentage of backfat loss during lactation compared to the control group, which could be attributed to the higher backfat thickness of sows in the treatment group prior to farrowing. It is well-established that backfat thickness before farrowing significantly influences the extent of backfat loss during lactation [9], which is in agreement with the findings of this study.

### 4.4. Piglet Colostrum Intake and Pre-Weaning Mortality

Dietary fiber supplementation did not affect piglet colostrum intake in this study. However, the percentages of inadequate colostrum intake in the control and treatment groups were high, at 16.9% and 17.9%, respectively. This might be attributed to the elevated pre-weaning mortality rate observed in piglets from both groups. The majority of piglet deaths, accounting for 66.7%, occurred within the first 3 days following birth and were correlated with the quantity of colostrum intake in the first 24 h of life. Piglets that consumed ≤300 g of colostrum had an excess mortality rate of 15.5%. In contrast, the pre-weaning mortality rate dropped to below 8.8% in piglets that consumed >300 g of colostrum. These results align with several studies that have highlighted the significance of colostrum intake for the survival of newborn piglets [22,33]. To enhance the piglet survival rate during lactation in modern sows, improving the colostrum intake of newborn piglets appears to be the most efficient strategy. Therefore, it is essential to eliminate factors that negatively impact sow colostrum production and those that diminish the vitality of newborn piglets.

### 4.5. Sow Colostrum Yield and Milk Yield

In the present study, Landrace × Yorkshire sows housed in a cooling system under tropical climates were found to produce approximately 5.26 kg of colostrum and between 7.60 and 9.22 kg of milk per day during the lactation periods of 3 to 10 days and 10 to 17 days, respectively. The milk production observed in this study was slightly lower than that reported in a previous study conducted in Thailand with Danish Landrace × Yorkshire crossbred sows, which showed milk production ranging from 9.3 to 11.1 kg per day during 3 to 10 days of lactation and from 12.3 to 13.1 kg per day during 10 to 17 days of lactation [9]. This difference could be attributed to the smaller litter sizes in the current study, with 13.8 TB and 11.8 BA, compared to 17.5 TB and 15.3 BA in the previous study [9]. The current study clearly demonstrates that TB significantly influences both colostrum yield and milk production in sows. Sows with larger litter sizes tend to produce more colostrum and milk than those with smaller litter sizes, possibly due to a more pronounced activation of the mammary glands immediately after delivery [34]. However, the study did not demonstrate any effect of dietary fiber supplementation on either colostrum yield or milk production.

## 5. Conclusions

Introducing dietary fiber to sows during the transition period reduced constipation and tended to decrease farrowing duration, particularly in sows with large litters. It also lowered pre-weaning piglet mortality by 3.9% in the free-farrowing system. However, providing sows with a 75 g/day dietary fiber supplement for only 7 days before farrowing proved insufficient to enhance colostrum and milk production and did not lead to increased colostrum consumption by piglets. Moreover, the treatment group exhibited greater backfat loss compared to the control group. As a result, it is advisable to carefully assess backfat levels prior to farrowing. For piglets, the amount of colostrum intake during the first 24 h after birth was found to be a critical factor in determining their pre-weaning mortality rate. Consequently, addressing the issue of constipation in peripartum sows and enhancing colostrum intake in newborn piglets emerge as pivotal strategies for improving piglet survival rates during lactation in modern prolific sow populations maintained in the free-farrowing system. However, additional studies are needed to explore the effects of higher fiber supplement doses and/or longer treatment periods during the late stages of gestation. This would provide sufficient time to enhance the intestinal microbiome of sows before farrowing.

## Figures and Tables

**Figure 1 animals-14-00854-f001:**
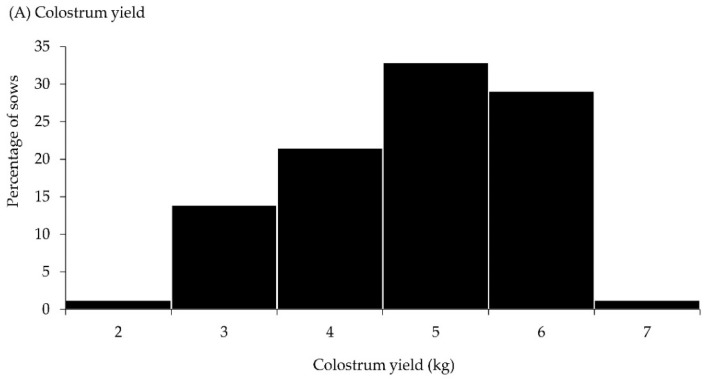
Frequency distribution of (**A**) colostrum yield (kg), (**B**) milk yield from 3–10 days (kg per day), and (**C**) milk yield from 10–17 days (kg per day) of lactation in Landrace × Yorkshire crossbred sows.

**Figure 2 animals-14-00854-f002:**
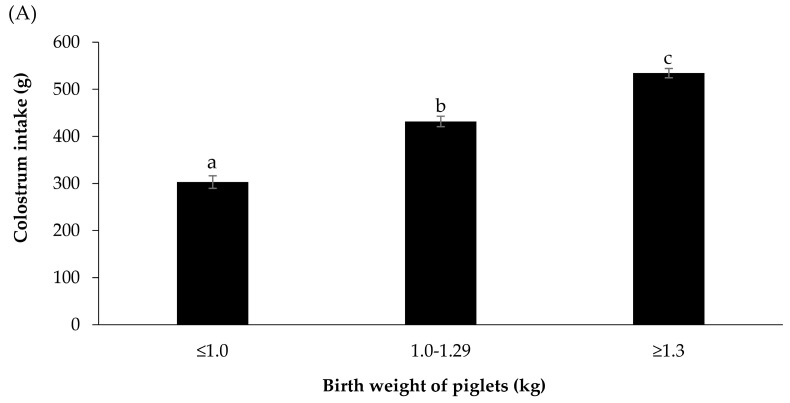
Colostrum intake (**A**) and the proportion of piglets with a colostrum intake <300 g (**B**) by birth weight of the piglet classes. a, b, and c different superscript letters indicate a significant difference (*p* < 0.001).

**Figure 3 animals-14-00854-f003:**
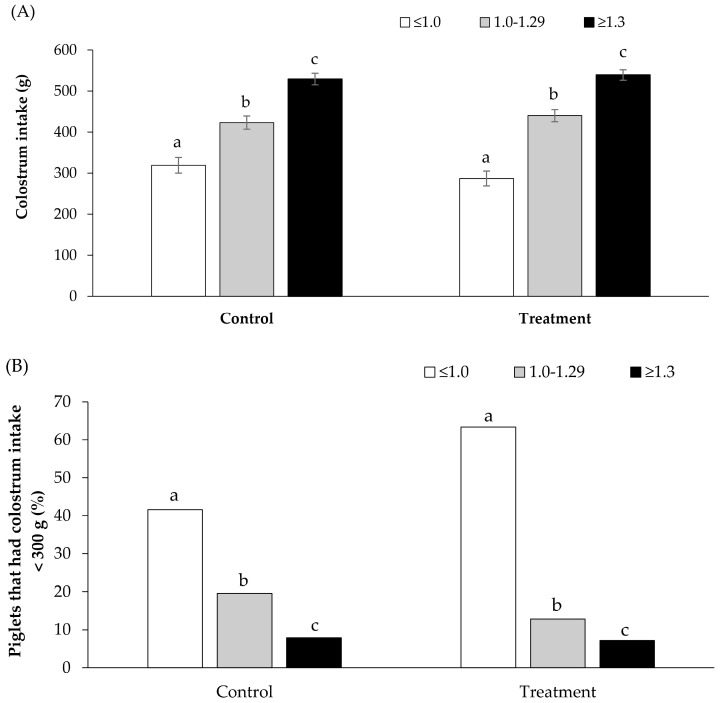
Colostrum intake (**A**) and the proportion of piglets with a colostrum intake < 300 g (**B**) by birth weight of the piglet classes in control and treatment groups. a, b, and c different superscript letters indicate a significant difference (*p* < 0.05).

**Figure 4 animals-14-00854-f004:**
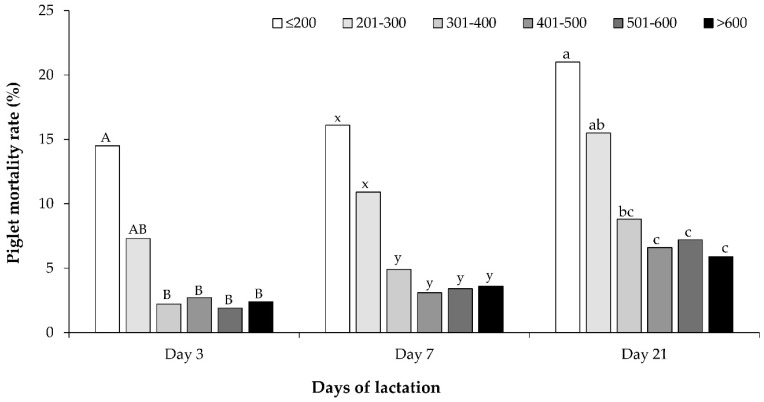
Percentages of piglet mortality within the first 3, 7, and 21 days of life categorized based on their colostrum intake during the first 24 h of life (≤200, 201–300, 301–400, 401–500, 501–600, and >600 g). A and B, x and y, and a, b, and c different superscripts indicated a significance difference within the same day of lactation (*p* < 0.05).

**Table 1 animals-14-00854-t001:** Composition of dietary fiber supplementation.

Chemical Composition (as Fed)	Percentage
Nutrients	
Crude protein	0.9
Crude fat	0.8
Crude fiber	59.0
Ash	1.0
Water	8.0
Fiber contents	
Total dietary fiber	88.0
Soluble fiber	1.3
Insoluble fiber	86.7
Neutral detergent fiber	78.0
Acid detergent fiber	64.0
Lignin	25.0
Bacterial fermentable substance	28.0

**Table 2 animals-14-00854-t002:** Composition of the control and experimental diets (as-fed basis).

Composition	Control Diet	Experimental Diet
Metabolizable energy (Kcal/kg)	3732	3732
Crude protein (%)	17.81	17.43
Crude fat (%)	5.94	5.82
Crude fiber (%)	4.30	5.53
Ash (%)	8.78	8.61
Lysine (%)	1.10	1.10

**Table 3 animals-14-00854-t003:** Descriptive statistics on the reproductive performance of Landrace × Yorkshire sows and the characteristics of their piglets.

Variables	Mean ± SD	Range
Sow (*n* = 92)		
Parity number	1.9 ± 0.7	1–3
Total number of piglets born per litter	13.8 ± 4.2	2–24
Number of piglets born alive per litter	11.8 ± 4.0	2–20
Stillborn piglets per litter (%)	7.4	0–42.8
Mummified fetuses per litter (%)	5.9	0–64.7
Farrowing duration (min)	216 ± 159	33–963
Sows with prolonged farrowing (%)	20.0	-
Colostrum yield (kg)	5.26 ± 1.08	2.40–7.85
Estimated colostrum IgG (%brix)	25.8 ± 3.2	18.8–33.5
Average daily feed intake (kg/sow/day)	3.9 ± 0.9	2.5–7.4
Milk yield during 3–10 days of lactation (kg/day)	7.60 ± 1.60	4.85–10.58
Milk yield during 10–17 days of lactation (kg/day)	9.22 ± 2.46	5.28–15.30
Backfat thickness at 109 days of gestation (mm)	21.6 ± 3.9	11.0–28.5
Backfat thickness at 21 days of lactation (mm)	14.8 ± 3.1	9.0–22.5
Backfat loss during lactation (%)	30.8 ± 13.0	−3.1–52.6
Piglets (*n* = 1272)		
Birth weight (g)	1330 ± 328	415–2150
Birth interval (min)	16.6 ± 46.8	0–894
Cumulative birth interval (min)	94.9 ± 98.8	0–963
Body weight at 24 h postpartum (g)	1451 ± 367	480–2400
Body weight gain during 0–24 h (g)	114.2 ± 115.	−725–730
Colostrum intake (g)	450.8 ± 160.9	19.2–954.6
Piglets that had colostrum intake < 300 g (%)	17.4	-
Piglet mortality rate during the first 3 days of life (%)	10.3	-
Piglet mortality rate during the first 7 days of life (%)	11.9	-
Piglet mortality rate during the first 21 days of life (%)	15.4	-

**Table 4 animals-14-00854-t004:** Reproductive performance, average daily feed intake, fecal score, colostrum yield, milk yield, and backfat thickness of sows from the control group (conventional lactation diet) and the treatment group (conventional lactation diet supplemented with fiber for 7 days before farrowing), presented as least-square means ± SEM.

Variables	Control	Treatment	*p* Value
Number of sows	41	51	-
Parity number	1.90 ± 0.10	1.94 ± 0.09	0.783
Total number of piglets born per litter	14.6 ± 0.5	14.7 ± 0.5	0.812
Number of piglets born alive per litter	12.4 ± 0.6	12.0 ± 0.5	0.661
Stillbirth (%)	7.2	9.5	0.273
Mummified fetuses (%)	7.2	7.9	0.828
Farrowing duration (min)	244.4 ± 30.7	200.8 ± 26.7	0.297
Sows with prolonged farrowing (%)	22.0	18.4	0.672
Average daily feed intake (kg/sow/day)	3.80 ± 0.18	3.91 ± 0.16	0.642
Fecal score (0–5)	2.08 ± 0.11	2.41 ± 0.09	0.019
Sows with constipation (%)	46.3	17.6	0.003
Brix value (%)	25.2 ± 0.6	26.1 ± 0.5	0.266
Colostrum yield (kg/day)	5.55 ± 0.21	5.05 ± 0.18	0.088
Milk yield during 3–10 days of lactation (kg/day)	7.85 ± 0.30	7.34 ± 0.27	0.212
Milk yield during 10–17 days of lactation (kg/day)	9.55 ± 0.47	8.76 ± 0.43	0.232
Backfat thickness at 109 days of gestation (mm)	19.6 ± 0.6	22.2 ± 0.6	0.003
Backfat thickness at 21 days of lactation (mm)	15.5 ± 0.5	14.1 ± 0.4	0.021
Backfat loss during lactation (%)	15.5 ± 0.5	34.8 ± 1.9	0.009

**Table 5 animals-14-00854-t005:** Piglet performance and colostrum intake for the control group (on a conventional lactation diet) and the treatment group (on a conventional lactation diet supplemented with fiber for 7 days before farrowing), presented as least-square means ± SEM.

Variables	Control	Treatment	*p* Value
Number of piglets	585	687	-
Birth weight (g)	1194 ± 15.3	1203 ± 13.9	0.649
Stillbirth (%)	7.4	10.3	
Birth interval (min)	11.2 ± 1.4	12.5 ± 1.4	0.527
Cumulative birth interval (min)	66.7 ± 9.6	87.1 ± 8.6	0.109
Body weight at 24 h postpartum (g)	1309 ± 23.0	1319 ± 13.6	0.733
Body weight gain during 0–24 h (g)	110.8 ± 10.8	115.9 ± 9.9	0.731
Colostrum intake (g)	424.0 ± 13.7	421.8 ± 12.6	0.908
Piglets that had a colostrum intake < 300 g (%)	16.9	17.9	0.688
Piglet mortality rate during the first 3 days of life (%)	8.9	11.5	0.172
Piglet mortality rate during the first 7 days of life (%)	11.2	12.5	0.513
Piglet mortality rate during the first 21 days of life (%)	17.3	13.4	0.085

## Data Availability

The data presented in this study are available on request from the corresponding author.

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
