# Peer review of "Impacts of Fiber Supplementation in Sows during the Transition Period on Constipation, Farrowing Duration, Colostrum Production, and Pre-Weaning Piglet Mortality in the Free-Farrowing System"

_animals, 2024, doi:10.3390/ani14060854_

Round 1

Reviewer 1 Report

Comments and Suggestions for Authors

I believe that the article was written on a current topic and is devoted to the current problem of maintaining the health of sows to produce healthy young pigs, increasing the safety of young animals, reducing the risks of complications during farrowing and after in a free farrowing. The system of group housing and free farrowing has a number of advantages and disadvantages. Optimizing the free-farrowing method in both European and tropical settings is essential to support sow and piglet welfare. The purpose of this study was to examine the impacts of dietary fiber supplementation during the transition period on constipation, farrowing du-ration, colostrum production, and stillbirth occurrence in sows, pre-weaning piglet mortality, litter weight gain, and sow milk yield.

The presented work is relevant both in fundamental and applied terms. To substantiate the relevance of the issue, references are given to modern authors on this topic. It was carried out at a good methodological level, using different methods. The zootechnical and statistical methods are described in great detail. To improve the quality of publication, I suggest the following:

1. The methodology should clarify how the supplement was given to the sows? At what time, how many times a day?

2. I would like to see the rationale for the dosage of the supplement? What were the authors guided by when determining the norm?

3. It would be desirable to indicate the feeding ration for experimental sows.

4. Figure 2 - reduce grams to g.

5. In conclusion, it is desirable to more clearly provide guidelines for future research in this direction.

I believe that this work can be published after corrections to minor methodological errors and text editing.

Author Response

# Reviewer 1

Comments and Suggestions for Authors

I believe that the article was written on a current topic and is devoted to the current problem of maintaining the health of sows to produce healthy young pigs, increasing the safety of young animals, reducing the risks of complications during farrowing and after in a free farrowing. The system of group housing and free farrowing has a number of advantages and disadvantages. Optimizing the free-farrowing method in both European and tropical settings is essential to support sow and piglet welfare. The purpose of this study was to examine the impacts of dietary fiber supplementation during the transition period on constipation, farrowing duration, colostrum production, and stillbirth occurrence in sows, pre-weaning piglet mortality, litter weight gain, and sow milk yield.

The presented work is relevant both in fundamental and applied terms. To substantiate the relevance of the issue, references are given to modern authors on this topic. It was carried out at a good methodological level, using different methods. The zootechnical and statistical methods are described in great detail. To improve the quality of publication, I suggest the following:

  1. The methodology should clarify how the supplement was given to the sows? At what time, how many times a day?

ANSWER: Additional data has been added: “The fiber supplement was provided in a dry form and mixed with the pellet-formed lactation feed before each feeding. All sows were manually fed four times a day at 6:00 a.m., 10:00 a.m., 1:00 p.m., and 4:00 p.m.”

  1. I would like to see the rationale for the dosage of the supplement? What were the authors guided by when determining the norm?

ANSWER: Additional rationale for the dose of the supplement has been addressed: “The amount of the dietary supplement given was determined by following the manufacturer's guidelines and taking into account the fiber and energy content of the standard feed. Usually, the purpose of giving dietary fiber to sows before farrowing is to enhance digestive health and alleviate constipation, a frequent issue during the final stages of pregnancy. The precise dosage may differ depending on the fiber type, the nutritional needs of the sow, and the complete diet plan. It is typical to incrementally raise the amount of fiber as farrowing nears, to smooth the transition and promote the health of both the sow and the soon-to-be-born piglets.”

  1. It would be desirable to indicate the feeding ration for experimental sows.

ANSWER: Feeding ration has been addressed “At various stages of gestation, the feed amount for each sow was adjusted daily. Initially, during the early, middle, and final stages of gestation, sows were fed 3.0–3.5 kg daily. The gestational feed comprised 12.7% crude protein, 2,700 kcal/kg of metaboliza-ble energy, 5.7% fiber, and 0.7% lysine. Upon moving to the farrowing house, sows switched to a lactation diet. In the week preceding farrowing, sows were gradually fed 3.0–3.5 kg of the lactation diet per day. This diet included 17.8% crude protein, 3,732 kcal/kg of metabolizable energy, 4.3% fiber, and 1.1% lysine. The calculated composition of the control and experimental diets in this study (as-fed basis) is presented in Table 2. Feeding occurred four times a day at set times: 06:00 AM, 10:00 AM, 01:00 PM, and 04:00 PM, with the feed presented in pellet form. Post-farrowing, sows had unlimited access to feed. An automatic feeder provided unrestricted access to the lactation diet for nursing sows, leading to an average intake of 5.0–6.0 kg per sow per day. Additionally, water was freely available via drinking nipples.”

  1. Figure 2 - reduce grams to g.

ANSWER: Adjusted as advised.

  1. In conclusion, it is desirable to more clearly provide guidelines for future research in this direction.

ANSWER: Additional conclusion has been provided: “……..However, additional studies are needed to explore the effects of higher fiber supplement doses and/or longer treatment periods during the late stages of gestation. This would provide sufficient time to enhance the intestinal microbiome of sows before farrowing”.

I believe that this work can be published after corrections to minor methodological errors and text editing.

ANSWER: Thank you very much.

Reviewer 2 Report

Comments and Suggestions for Authors

Dear authors,

Your paper "Impacts of Fiber Supplementation in Sows During the Transition Period on Constipation, Farrowing Duration, Colostrum Production, and Pre-Weaning Piglet Mortality in the Free-Farrowing System" it is very interesting and for sure it shows a lot of work and study involved. I would like to ask you to add the chemical composition of the two tested ingredients starch fibre and konjac flour, and also the chemical composition of control, and experimental diets ). Thank you.

Simple Summary:  please remove dietary or diet, use one or another

Introduction: It is well-written and comprehensive, explain the background and the state of  art of the farrowing phase in sows  the chemical composition of the two tested ingredients starch fibre and konjac flour, and also the chemical composition of control, and experimental diets )

Material and Methods

Row 128: Please mention the date of protocol no. 233124 when it was approved/issued

Row 180: Can you add more info about how you provide water? Was it through nipple systems?

Results

Mandatory

Also add the chemical composition of two main fiber sources : starch fibre and konjac flour

 Please add the diet ingredients, chemical composition for control, and experimental diets.

Author Response

# Reviewer 2

Comments and Suggestions for Authors

Dear authors,

Your paper "Impacts of Fiber Supplementation in Sows During the Transition Period on Constipation, Farrowing Duration, Colostrum Production, and Pre-Weaning Piglet Mortality in the Free-Farrowing System" it is very interesting and for sure it shows a lot of work and study involved. I would like to ask you to add the chemical composition of the two tested ingredients starch fibre and konjac flour, and also the chemical composition of control, and experimental diets ). Thank you.

ANSWER: Additional data on composition of the tested fiber has been provided (see Table 1).

Simple Summary:  please remove dietary or diet, use one or another

ANSWER: Removed as suggested.

Introduction: It is well-written and comprehensive, explain the background and the state of art of the farrowing phase in sows the chemical composition of the two tested ingredients starch fibre and konjac flour, and also the chemical composition of control, and experimental diets).

ANSWER: Additional information has been added: “Earlier research indicates that konjac flour is primarily composed of glucomannan, a polymer connected by β-1,4-glycosidic bonds [18]. In contrast, resistant starch, mainly consisting of type 3 resistant starch, is produced through the gelatinization and subsequent recrystallization of amylose and amylopectin. In vitro fermentation experiments revealed that resistant starch contains higher levels of formic acid and lactate compared to konjac flour, which, conversely, exhibits higher concentrations of propionate and butyrate. Additionally, the fermentation studies showed that the populations of Anaerovibrio and Erysipelatoclostridium were more abundant in konjac flour fermentations, whereas Proteiniclasticum was more prevalent in those involving resistant starch [18]. These findings suggest that different fiber sources can distinctly influence the intestinal microbiome. Nonetheless, further research is necessary to explore the clinical implications of various fiber types on constipation and sow reproductive performance.”

Material and Methods

Row 128: Please mention the date of protocol no. 233124 when it was approved/issued

ANSWER: Additional information was provided: “It was authorized ….… (protocol number 233124, approved July 1, 2023).”

Row 180: Can you add more info about how you provide water? Was it through nipple systems?

ANSWER: Additional information was provided: “Additionally, water was freely available via drinking nipples. (L. 210)”

Results

Mandatory

Also add the chemical composition of two main fiber sources : starch fibre and konjac flour

ANSWER: Further details on resistant starch and konjac flour were discussed in the introduction section. However, these two fiber types were not utilized in our experiment. The chemical composition of the fiber employed in this study is detailed in Table 1.

Please add the diet ingredients, chemical composition for control, and experimental diets.

ANSWER: Additional information concerning the ingredients composition of control and experimental diet has been provided in Table 2.

Table 2. Composition of the control and experimental diets (as-fed basis).

Composition

Control diet

Experimental diet

Metabolizable energy (Kcal/kg)

3732

3732

Crude protein (%)

17.81

17.43

Crude fat (%)

5.94

5.82

Crude fiber (%)

4.30

5.53

Ash (%)

8.78

8.61

Lysine (%)

1.10

1.10

Reviewer 3 Report

Comments and Suggestions for Authors

The manuscript presents a thorough investigation into the effects of dietary fiber supplementation on the health and performance of sows and their piglets. The study meticulously explores various outcomes such as constipation prevalence in sows, farrowing duration, colostrum and milk yield, and piglet pre-weaning mortality. Utilizing a substantial sample size and a controlled experimental design, the paper offers valuable insights, suggesting that dietary fiber supplementation can significantly impact sow well-being and potentially enhance piglet survival rates.

Comments:

Line 51. Where do the mortality rates come from? Control or Treatment group?

Line 137. Although there is plenty of information described in the method part, the formulation should be provided as a table in the method section.

Line 146. It is valuable that the author provided detailed information about animal experiment and management. However, the sample and data collection part might be separated from the Animal and Experimental Design. A separated subtitle (For example, 2.2 Sow characteristics; 2.3 Piglet characteristics) about each parameter and how did the author collect them would be clearer and easier for reader to understand. 

Line 253. The GLIMMIX were mentioned here. However, the data was not showed in the result parts. It seems like just the Mean and P values were mentioned in the results.

Line 276. Why the author chose to present the data in two groups together? Why not compare different groups? What is the idea to combine the data of Control and Treatment together?

Line 290. In the Table 1, the data was shown as Means ± SD, but in Table 2 was Means ± SEM. This should be in consistence and described in the Method section.

Tables. Each table should have a Legend to clearly describe the data.

Line 280. What is the duration of the feed intake data been collected?

Author Response

# Reviewer 3

The manuscript presents a thorough investigation into the effects of dietary fiber supplementation on the health and performance of sows and their piglets. The study meticulously explores various outcomes such as constipation prevalence in sows, farrowing duration, colostrum and milk yield, and piglet pre-weaning mortality. Utilizing a substantial sample size and a controlled experimental design, the paper offers valuable insights, suggesting that dietary fiber supplementation can significantly impact sow well-being and potentially enhance piglet survival rates.

Comments:

Line 51. Where do the mortality rates come from? Control or Treatment group?

ANSWER: The mortality rate mentioned in the abstract represents an average from both the control and treatment groups. The sentence has been modified to “Across groups, piglet mortality rates within 3, 7, and 21 days were 10.3%, 11.9%, and 15.4%, respectively.”

Line 137. Although there is plenty of information described in the method part, the formulation should be provided as a table in the method section.

ANSWER: The summarized tables of chemical composition of traditional lactation diet and fiber supplementation has been added (Table 1).

Line 146. It is valuable that the author provided detailed information about animal experiment and management. However, the sample and data collection part might be separated from the Animal and Experimental Design. A separated subtitle (For example, 2.2 Sow characteristics; 2.3 Piglet characteristics) about each parameter and how did the author collect them would be clearer and easier for reader to understand.

ANSWER: Data collection part has been separated to 2.2 Sow characteristics and 2.3 Piglet characteristics as suggested.

Line 253. The GLIMMIX were mentioned here. However, the data was not showed in the result parts. It seems like just the Mean and P values were mentioned in the results.

ANSWER: The data using GLIMMIX analysis, comparing sows with prolonged farrowing and sows with constipation between control and treatment groups, have shown in Table 3.

Line 276. Why the author chose to present the data in two groups together? Why not compare different groups? What is the idea to combine the data of Control and Treatment together?

ANSWER: Additional statement concerning GLIMMIX analysis has been added: “For categorical data, including the proportion of piglets that were born with SB birth status (yes/no), the proportion of piglets with colostrum intake below 300 g (yes/no), and the percentage of piglet mortality at 3, 7, and 21 days of lactation, analysis was conducted using the GLIMMIX procedure. The statistical models included dietary fiber supplementation (control and treatment), sow parity (primiparous and multiparous), piglet birth weight categories (<1.0 kg, 1.0–1.29 kg, and ≥1.3 kg), and the interactions between treatment groups and sow parity, as well as between treatment groups and piglet birth weight categories as fixed effects. To adjust for repeated measurements, the sow's identity was included as a random effect in the model. Least square means were calculated for each variable and subsequently compared using the least significant difference test.”

Line 290. In the Table 1, the data was shown as Means ± SD, but in Table 2 was Means ± SEM. This should be in consistence and described in the Method section.

ANSWER: Descriptive statistics were performed for all analyzed traits and are succinctly presented in Table 1 as Means ± SD, along with the data range. For statistical comparisons presented in Tables 3, 4, and 5, least-square means ± SEM were utilized. Detailed explanations of these methodologies are provided in the 'Statistical Analysis' section.

Tables. Each table should have a Legend to clearly describe the data.

ANSWER: Legend description for each table has been added.

Line 280. What is the duration of the feed intake data been collected?

ANSWER: Additional information has been added in M&M: “Two hours post-feeding, any uneaten feed was gathered, weighed, and discarded. The consumption for each meal was calculated by subtracting the amount of remaining feed from the initial quantity provided to the sows.” Thank you very much.

Round 2

Reviewer 2 Report

Comments and Suggestions for Authors

Dear authors,

Overall, the manuscript was well improved and you have taken into consideration the suggestions I made,  therefore I recommend it for publication. 

Please mind the extra dash  from ABSTRACT, row 53.

Also, do not leave a space  on Table 5. Add a dash"-"

P

Reviewer 3 Report

Comments and Suggestions for Authors

The author has answered all my questions.